# Contrastive Gradient Guidance for Test-time Preference Alignment of Diffusion Models

## Abstract

Pre-trained diffusion models demonstrate remarkable performance in text-to-image generation, with current research efforts directed toward aligning them with human preferences across diverse application scenarios. Existing approaches often rely on costly pipelines that require collecting preference data, training reward models, and fine-tuning. A promising alternative is test-time alignment, which steers diffusion models during sampling without retraining. However, current test-time alignment methods typically depend on explicit reward models to provide a guidance signal for modifying a sampling path. These involve decoding a noisy image and estimating its rewards, which adds extra steps with computational overhead and might limit flexibility across diverse scenarios. We propose Contrastive Gradient Guidance (CGG), a conceptually straightforward and practical framework for test-time alignment that avoids explicit reward models by design. CGG derives its guidance signal from the contrastive difference between two diffusion models, parameterized through the gradient of the log-likelihood ratio of the favored and the unfavored distributions. The guidance signal steers a pre-trained diffusion model along its sampling path while implicitly aligning generation with human preferences. Experiments demonstrate that CGG consistently improves preference alignment in text-to-image generation and flexibly adapts to safety-critical and multi-preference scenarios. Moreover, CGG can be combined with prevailing test-time alignment techniques to yield additional gains. These results establish CGG as a principled framework for advancing test-time alignment of diffusion models.[1][2]

## 1 Introduction

Pre-trained diffusion models have emerged as a powerful class of generative models, producing high-quality, diverse outputs across various domains. The generation paradigm has shown promise in image generation (Podell et al., 2024), video generation (Ho et al., 2022), and speech generation (Huang et al., 2022). However, steering these models to generate content aligned with specific user requirements or robust quality remains challenging, which is known as the preference alignment problem. The preference alignment problem aims to ensure that the model-generated results match desirable goals (Ngo et al., 2022).

Existing approaches often rely on costly pipelines that require collecting preference data, training reward models (for human preferences), and fine-tuning. For example, Christiano et al. (2017) presented reinforcement learning from human feedback (RLHF), which fine-tunes a generative model by a reward model learned from a pairwise preference dataset. This work laid the foundation for the RLHF training paradigm of generative models. Alternatively, Rafailov et al. (2023) proposed a supervised alternative of RLHF–direct preference optimization (DPO), which fine-tunes a generative model from a pairwise preference dataset without training a reward model. DPO simplifies RLHF by bypassing the training of an explicit reward model and directly deriving an objective function to fine-tune the diffusion model. This family of methods, including $f$-DPO (Wang et al., 2023), KTO (Ethayarajh et al., 2024), SimPO (Meng et al., 2024), and ORPO (Hong et al., 2024a) demonstrate competitive performance to RLHF.

---

[1] **Warning: This paper contains examples of harmful content, including explicit text and images.**
[2] This work use Large Language Models (LLMs) in paper writing to aid or polish writing.

Although the RLHF and DPO appear promising for the preference alignment during the training time, these fine-tuning method requires expensive training resources for hyper-parameter tuning and are inflexible for adapting a diffusion model to new human preferences, such as personalization or safety concerns. A promising alternative is test-time preference alignment, which steers diffusion models during sampling without retraining. For example, *prompt optimization* is an early proposed method for low-cost test-time preference alignment, which optimizes the input text prompt automatically to get generation quality (Hao et al., 2023; Mo et al., 2024). Similarly, *initial noise optimization* focuses on finding the good initial noisy inputs to generate high-quality information (Guo et al., 2024). The core idea of these methods is to optimize inputs and internal mechanisms of text-to-image diffusion models for test-time preference alignment (Liu et al., 2024). While several works put efforts into optimizing the inputs for test-time preference alignment, the control of generating results is less precise and often relies on heuristics.

In this work, we focus on the *gradient guidance* techniques of diffusion models for test-time preference alignment, where the main idea is to directly sample from the output distribution by introducing the guidance signals to the denoising process. Existing methods typically rely on guidance signals from explicit reward models to modify sampling paths. However, these methods involve decoding the noisy images from the latent space, estimating the expected reward of a clean image estimated from a noisy image, and designing the gradient guidance signal (Chung et al., 2022; Bansal et al., 2023; Yu et al., 2023) or the re-sampling method, e.g., Sequential Monte Carlo (SMC) Sampling (Wu et al., 2023; Kim et al., 2025; Singhal et al., 2025).

In this work, we propose Contrastive Guidance Generation (CGG), which enables effective test-time preference alignment for diffusion models **without** explicit reward models by default. CGG derives its guidance signal from the contrastive difference between two diffusion models, parameterized through formalizing the reward model as the log-likelihood ratio of the favored and the unfavored distributions. Based on this interpretation, CGG leverages classifier guidance techniques with the guidance signal to steer a pre-trained diffusion model along its sampling path for image quality and diversity while implicitly toward preference-aligned outputs.

We demonstrate that CGG consistently improves a pre-trained diffusion model's preference alignment performance in text-to-image generation during test time by showing the state-of-the-art performance evaluated by PickScore on a Pick-a-Pic test set. We also present the flexibility of CGG by utilizing different compositions of diffusion models to adapt a pre-trained diffusion model to safety-critical (Pick-Safety) and multiple preferences (PickScore and ImageReward).

Our contribution is to propose the Contrastive Gradient Guidance (CGG) framework for test-time preference alignment that avoids explicit reward models by design. We demonstrate that CGG consistently improves the pre-trained diffusion model and can be extended to diverse scenarios. Our framework can connect with current research on diffusion models in preference alignment, providing a new perspective for test-time preference alignment problems.

## 2 BACKGROUND

### 2.1 CLASSIFIER GUIDANCE FOR DIFFUSION MODEL

Instead of estimating a data distribution $p(\boldsymbol{x})$, diffusion models formulate the generative process as the iterative denoising process from the tractable Gaussian distribution, which estimates the score of the distribution at each iteration $t$ by the denoising model $s_\theta(\boldsymbol{x}_t) \to \nabla_{\boldsymbol{x}_t} \log p(\boldsymbol{x}_t)$ (Song et al., 2021b). Conditional generation further enhances its generation controllability by incorporating conditions into the denoising process. For example, classifier guidance (CG) trains a time-dependent classifier $p(\boldsymbol{y}|\boldsymbol{x}_t)$ on a noisy image $\boldsymbol{x}_t$ to adjust the unconditional by its gradient based on Bayes' rule (Dhariwal & Nichol, 2021; Song et al., 2021b).

$$\nabla_{\boldsymbol{x}_t} \log p(\boldsymbol{x}_t|\boldsymbol{y}) = \underbrace{\nabla_{\boldsymbol{x}_t} \log p(\boldsymbol{x}_t)}_{\text{unconditional score}} + \underbrace{\nabla_{\boldsymbol{x}_t} p(\boldsymbol{y}|\boldsymbol{x}_t)}_{\text{classifier/conditional gradient}} - \underbrace{\nabla_{\boldsymbol{x}_t} \log p(\boldsymbol{y})}_{=0}, \tag{1}$$

Therefore, the classifier gradient $\nabla_{\boldsymbol{x}_t} p(y|\boldsymbol{x}_t)$ acts as the guidance signal to steer the model to align the conditions, such as a class label or a text sequence.

## 2.2 RLHF FOR TRAINING-TIME PREFERENCE ALIGNMENT

In RLHF, we collect pairwise preference data from human annotators, which is a set of pairs of images $(\boldsymbol{x}^w, \boldsymbol{x}^l)$ with a prompt $\boldsymbol{c}$ in the context of the text-to-image scenario, where $\boldsymbol{x}^w$ is the favored image and $\boldsymbol{x}^l$ is the less favored image. The first stage of RLHF is reward modeling, which learns a distribution over the pairwise preference data to represent human preferences by modeling them as a Bradley-Terry model (Bradley & Terry, 1952). The Bradley-Terry model assumes that the probability of one image being favored over another can be expressed as a reward function of their respective rewards:

$$\mathbb{P}\left[y = +1 | (\boldsymbol{c}, \boldsymbol{x}^w, \boldsymbol{x}^l) \in \mathcal{D}\right] = \sigma\left(r(\boldsymbol{c}, \boldsymbol{x}^w) - r(\boldsymbol{c}, \boldsymbol{x}^l)\right), \tag{2}$$

where $\sigma(\bullet)$ is the sigmoid function and $y = \mathbb{1}[\boldsymbol{x}^w \succ \boldsymbol{x}^l | \boldsymbol{c}]$ is the label of a pair of images given a certain prompt, $(\boldsymbol{c}, \boldsymbol{x}^w, \boldsymbol{x}^l) \in \mathcal{D}$ represents a prompt and a pair of images, and $r(\boldsymbol{c}, \boldsymbol{x})$ is a reward function that assigns a score to each image. As Eq. 2 illustrates, the reward modeling could be formulated as a binary classification problem with the following negative log-likelihood problem:

$$\min_{\phi} - \log\left(\sigma\left(r_{\phi}(\boldsymbol{c}, \boldsymbol{x}^w) - r_{\phi}(\boldsymbol{c}, \boldsymbol{x}^l)\right)\right). \tag{3}$$

In the second stage of RLHF (proximal policy optimization), we formulate the objective function as maximizing a generative model to get a high reward score $r(\boldsymbol{c}, \boldsymbol{x})$ while penalizing the KL divergence between the model distribution and the reference distribution. The objective function can be expressed as:

$$\max_{\theta} \mathbb{E}_{\boldsymbol{x} \sim p_{\theta}(\boldsymbol{x}|\boldsymbol{c})}\left[r(\boldsymbol{c}, \boldsymbol{x})\right] - \beta \mathbb{D}_{\mathrm{KL}}[p_{\theta}(\boldsymbol{x}|\boldsymbol{c}) \| p_{\mathrm{ref}}(\boldsymbol{x}|\boldsymbol{c})], \tag{4}$$

where $p_{\theta}(\boldsymbol{x}|\boldsymbol{c})$ is the distribution of the fine-tuned generative model, $\beta$ is a hyperparameter to scale the regularization term. Given a reward model $r(\boldsymbol{c}, \boldsymbol{x})$ and a $\beta$, Rafailov et al. (2023) derive the optimal solution of the objective as:

$$p_{\theta^*}(\boldsymbol{x}|\boldsymbol{c}) = \frac{1}{Z(\boldsymbol{c})} p_{\mathrm{ref}}(\boldsymbol{x}|\boldsymbol{c}) \exp\left(\frac{1}{\beta} r(\boldsymbol{c}, \boldsymbol{x})\right), \tag{5}$$

where $Z(\boldsymbol{c}) = \int_{\boldsymbol{x}} p_{\mathrm{ref}}(\boldsymbol{x}|\boldsymbol{c}) \exp\left(\frac{1}{\beta} r(\boldsymbol{c}, \boldsymbol{x})\right)$ is the partition function.

# 3 CONTRASTIVE GRADIENT GUIDANCE

In this work, we investigate a method to effectively and flexibly sample an image $\boldsymbol{x}$ from the preference-aligned target distribution $p_{\theta^*}(\boldsymbol{x}|\boldsymbol{c})$ in Eq. 5 during the test time.

## 3.1 PROBLEM FORMULATION AND MOTIVATION

**Test-time preference alignment.** Given a pre-trained diffusion model $s_{\mathrm{ref}}(\boldsymbol{c}, \boldsymbol{x})$, reward model $r(\boldsymbol{c}, \boldsymbol{x})$, and the test set prompt $\{\boldsymbol{c}\}$, the goal is to adjust the output distribution of a pre-trained diffusion model without modifying its parameters.

As mentioned in Section 2.1, we can estimate a conditional distribution $p_{\theta^*}(\boldsymbol{x}|\boldsymbol{c})$ by its score function like Eq. 6:

$$\nabla_{\boldsymbol{x}_t} \log p_{\theta^*}(\boldsymbol{x}_t|\boldsymbol{c}) = \nabla_{\boldsymbol{x}_t} \log p_{\mathrm{ref}}(\boldsymbol{x}_t|\boldsymbol{c}) + \gamma \nabla_{\boldsymbol{x}_t} r(\boldsymbol{c}, \boldsymbol{x}_t), \tag{6}$$

where $\gamma$ is the guidance scale; $\nabla_{\boldsymbol{x}_t} \log p_{\mathrm{ref}}(\boldsymbol{x}_t|\boldsymbol{c})$ is the score of the pre-trained distribution at timestep $t$ corresponding to the pre-trained diffusion model $s_{\mathrm{ref}}(\boldsymbol{c}, \boldsymbol{x}_t)$ and $\nabla_{\boldsymbol{x}_t} r(\boldsymbol{c}, \boldsymbol{x}_t)$ is the gradient of the reward on a generated image, denoted as the reward guidance signal. Eq. 6 suggests that if we can accurately estimate the reward guidance signal $\nabla_{\boldsymbol{x}_t} r(\boldsymbol{c}, \boldsymbol{x}_t)$, we can sample an image $\boldsymbol{x}$ from the preference-aligned target distribution.

Naturally, Eq. 1 tells us that it is possible to train a time-dependent reward model to get the reward guidance signal. However, training a time-dependent reward model requires collecting a large set of noisy data with rewards $(\boldsymbol{c}, \boldsymbol{x}_t)$ (Singhal et al., 2025). In our ablation study (Appendix A.), we demonstrate that fine-tuning *time-dependent reward model* from the reward model $r(\boldsymbol{c}, \boldsymbol{x}_0)$ on pairwise preference data $\{(\boldsymbol{c}, \boldsymbol{x}_t^w, \boldsymbol{x}_t^l)\}$ is unstable and costly.

To tackle the problem of estimating the reward guidance signal $\nabla_{\boldsymbol{x}_t} r(\boldsymbol{c}, \boldsymbol{x}_t)$ during the test time, two types of explicit reward-guided methods have been proposed: gradient-free steering and gradient guidance. Both method estimates a reward of an estimated clean image from a noisy image $\hat{\boldsymbol{x}}(\boldsymbol{x}_t) = \mathbb{E}[\boldsymbol{x}|\boldsymbol{x}_t]$. Gradient-free steering utilizes the estimated reward to perform the Sequential Monte Carlo (SMC) sampling for increasing samples' rewards during the test time (Wu et al., 2023; Kim et al., 2025; Singhal et al., 2025), which is orthogonal to our approach, and we leave details in Section 5. Gradient guidance calculates the gradient of the estimated reward as the reward guidance signal $\nabla_{\boldsymbol{x}_t} r(\boldsymbol{c}, \hat{\boldsymbol{x}}(\boldsymbol{x}_t))$ for gradient guidance during sampling (Chung et al., 2022; Bansal et al., 2023; Yu et al., 2023). However, these approaches have studied the specific formulation of the reward model for image generation, such as the inverse problem (Chung et al., 2022) or classification (Yu et al., 2023; Bansal et al., 2023), instead of the pairwise preference alignment tasks as mentioned in Section 2.2.

## 3.2 CONTRASTIVE FORM AND GUIDANCE

Existing works rely on an external reward model to provide the reward guidance signal for guiding. In this work, we ask the question about

*Is there an approximation of the reward guidance signal without an external reward model?*

We begin by the Eq. 5:

$$
\begin{aligned}
\exp\left(\frac{1}{\beta}r(\boldsymbol{c}, \boldsymbol{x})\right) &= Z(\boldsymbol{c})\frac{p_{\theta^*}(\boldsymbol{x}|\boldsymbol{c})}{p_{\text{ref}}(\boldsymbol{x}|\boldsymbol{c})} \\
\frac{1}{\beta}r(\boldsymbol{c}, \boldsymbol{x}) &= \log\left(Z(\boldsymbol{c})\frac{p_{\theta^*}(\boldsymbol{x}|\boldsymbol{c})}{p_{\text{ref}}(\boldsymbol{x}|\boldsymbol{c})}\right).
\end{aligned}
\tag{7}
$$

Thus, maximizing the rewards in RLHF with the KL divergence could be framed as the optimization problem of maximizing the distinction between distributions. To maximize the distinction between distributions during the test time, we propose the **contrastive form** as the reward guidance signal would be tractable and achieve satisfying outcomes. Specifically, we argue that the **contrastive form** expresses that a reward model during the test time is proportional to the contrastive difference between the favored and the unfavored distributions, parameterized through the log-likelihood ratio:

$$
r(\boldsymbol{c}, \boldsymbol{x}) \propto \log\frac{\rho(\boldsymbol{x}|\boldsymbol{c})}{\kappa(\boldsymbol{x}|\boldsymbol{c})},
\tag{8}
$$

where $\rho(\boldsymbol{x}|\boldsymbol{c})$ and $\kappa(\boldsymbol{x}|\boldsymbol{c})$ represent the favored and the unfavored distributions. Intuitively, images collected from $\rho(\boldsymbol{x}|\boldsymbol{c})$ represent high-reward samples, whereas those from $\kappa(\boldsymbol{x}|\boldsymbol{c})$ represent low-reward samples, which indicates that the probability of image $\boldsymbol{x}$ with a higher reward under the favored distribution is expected to be higher than under the unfavored distribution for a sample prompt $\boldsymbol{c}$. In RLHF, the distribution modeling by the preference optimization (PO)-based is naturally treated as the favored distribution, and the pre-trained distribution is the unfavored distribution. In addition, we argue that Eq. 8 is flexible by discussing other kinds of contrastive forms in Section 3.3.

Consequently, we build the Contrastive Gradient Guidance (CGG) framework, which estimates the preference-aligned target distribution $p_{\theta^*}(\boldsymbol{x}|\boldsymbol{c})$ by replacing the $\nabla_{\boldsymbol{x}} r(\boldsymbol{c}, \boldsymbol{x})$ with the contrastive form in Eq. 6:

$$
\begin{aligned}
\nabla_{\boldsymbol{x}_t} \log p_{\theta^*}(\boldsymbol{x}_t|\boldsymbol{c}) &= \nabla_{\boldsymbol{x}_t} \log p_{\text{ref}}(\boldsymbol{x}_t|\boldsymbol{c}) + \gamma\nabla_{\boldsymbol{x}_t}\left(\log\frac{\rho(\boldsymbol{x}_t|\boldsymbol{c})}{\kappa(\boldsymbol{x}_t|\boldsymbol{c})} + C\right) \\
&= \nabla_{\boldsymbol{x}_t} \log p_{\text{ref}}(\boldsymbol{x}_t|\boldsymbol{c}) + \gamma\left(\nabla_{\boldsymbol{x}_t} \log \rho(\boldsymbol{x}_t|\boldsymbol{c}) - \nabla_{\boldsymbol{x}_t} \log \kappa(\boldsymbol{x}_t|\boldsymbol{c})\right), \\
&\leftarrow s_{\text{ref}}(\boldsymbol{c}, \boldsymbol{x}_t) + \gamma(s_{\theta_\rho}(\boldsymbol{c}, \boldsymbol{x}_t) - s_{\theta_\kappa}(\boldsymbol{c}, \boldsymbol{x}_t)),
\end{aligned}
\tag{9}
$$

where $C$ is the constant and $\gamma$ is the guidance scale for the test-time preference alignment. Notably, we introduce two diffusion models $s_{\theta_\rho}(\boldsymbol{c}, \boldsymbol{x}_t), s_{\theta_\kappa}(\boldsymbol{c}, \boldsymbol{x}_t)$ to estimate the scores of favored and unfavored distributions in Eq. 9.

In Section 3.3, we describe different types of combinations of two diffusion models under the CGG framework.

**Remark of CGG for test-time preference alignment.** CGG is a straightforward and practical framework for test-time preference alignment, which could be a good replacement for training a time-dependent reward model. In addition, comparing with existing explicit reward-guided methods (Chung et al., 2022; Bansal et al., 2023; Yu et al., 2023; Yeh et al., 2025; Wu et al., 2023; Kim et al., 2025; Singhal et al., 2025), CGG integrates an implicit reward guidance signal into the pre-trained diffusion model's sampling path, which does not rely on an external reward model. We demonstrate the effectiveness and flexibility of the CGG framework in Section 4 and discuss our method with existing test-time preference alignment in Section 5.

### 3.3 IMPLEMENTATION OF THE CONTRASTIVE FORM

**Preference optimization fine-tuned and pre-trained diffusion models.** In this work, we reuse the preference optimization (PO) fine-tuned diffusion models such as Diffusion-DPO and Diffusion-KTO as the $s_{\theta_\rho}(c, x_t) = s_{\text{PO}}(c, x_t)$ to estimate the score of the favored distribution. In particular, we take the pre-trained diffusion model as the estimation of unfavored score $s_{\theta_\kappa}(c, x_t) = s_{\text{ref}}(c, x_t)$ and derive the form:

$$\nabla_{x_t} \log p_{\theta^*}(x_t|c) \leftarrow s_{\text{ref}}(c, x_t) + \gamma(s_{\text{PO}}(c, x_t) - s_{\text{ref}}(c, x_t))$$
$$= (1 - \gamma)s_{\text{ref}}(c, x_t) + \gamma s_{\text{PO}}(c, x_t) \tag{10}$$

Eq. 10 connects the CGG framework with a broadly used classifier-free guidance (CFG) (Ho & Salimans, 2022) to push the sampling path towards higher likelihood satisfying conditions for conditional guided sampling. In subsequent work, Karras et al. (2024) verifies that the image quality could be further improved by applying CFG to a high-quality diffusion model with a poor diffusion model, both trained on the same task, conditioning, and data distribution. In Section 4, our experimental results show that adopting CGG with this certain form, we could additionally enhance the rewards score by guide the pre-trained diffusion model with a PO fine-tuned diffusion model.

**Fine-tuned two diffusion models from pairwise preference data.** We also explore the alternative way to estimate the reward signal by the other composition of the **contrastive form**. In practice, a pairwise preference dataset is collected for RLHF, e.g., a Pick-a-Pic format is $\mathcal{D} = \{(c, x^{(0)}, x^{(1)}, y)\}$, which contains a pair of images $(x^{(0)}, x^1)$ for each prompt $c$ and preference label $y$. For this kind of pairwise preference dataset, the first idea is to utilize the Supervised Fine-Tuning (SFT) method on the positive label and the negative label. Thus, we could obtain the score models of the positive label distribution (favored distribution) $s_{\theta_\rho}(c, x_t) = s_{\text{Pos}}(c, x_t)$ and the negative label distribution (unfavored distribution) $s_{\theta_\kappa}(c, x_t) = s_{\text{Neg}}(c, x_t)$.

A similar idea is proposed by the CHATS, which claimed that the DPO's objective function does not meet the properties of Classifier-Free Guidance (CFG) to generate high quality (rewards) and diverse images by only one diffusion model, so they proposed the new fine-tuning objective function to train the two diffusion models for sampling from the favored and the unfavored distributions simultaneously (Fu et al., 2025). In our implementation, we utilize CHATS's favored diffusion models $s_{\theta_\rho}(c, x_t) = s_{\text{CHATS}+}(c, x_t)$ and unfavored diffusion models $s_{\theta_\kappa}(c, x_t) = s_{\text{CHATS}-}(c, x_t)$ to form the **contrastive form**.

**Multiple diffusion models for multiple preferences.** Eq. 9 illustrates that the score of the preference-aligned target distribution $\nabla_{x_t} \log p_{\theta^*}(x_t|c)$ with respect to a single reward model $r(c, x_t)$ could be approximate by the **contrastive form**. This form inspires us to ask the next question

*Would the multiple preference-aligned target distribution be estimated by reward guidance signals?*

Previous works provide the potential of this idea by mapping multiple conditions (concepts) to constitute generated images (Liu et al., 2022). Based on a similar idea, we extend Eq. 9 to apply multiple preferences composed of multiple diffusion models fine-tuned from different pairwise preference datasets.

$$\nabla_{x_t} \log p_{\Theta^*}(x_t|c) \leftarrow s_{\text{ref}}(c, x_t) + \sum_{k=1}^{K} \gamma_k(s_{\theta_{\rho,k}}(c, x_t) - s_{\theta_{\kappa,k}}(c, x_t)), \tag{11}$$

Table 1: Comparison of average rewards between SDXL, Diffusion-DPO and applying CGG in DPO using prompts from the Pick-a-Pic v2 test set. We use $\gamma* = 0.75$. **Bold text** results represents the best among experiments. Additionally, $\mathrm{CGG}(s_{\theta_\rho}(c, x_t), s_{\theta_\kappa}(c, x_t))$ represents the different composition of the contrastive form.

|  | SDXL(Ref) | DPO | CGG(DPO, Ref) ($\gamma^*$=0.75) |
|---|---|---|---|
| Pickscore | 22.114 | 22.408 | **22.481** |
| Aesthetic | **6.481** | 6.437 | 6.449 |
| HPSv2 | 0.292 | 0.303 | **0.305** |
| CLIP | 36.994 | 37.783 | **37.978** |
| ImageReward | 0.857 | 0.991 | **1.040** |

Table 2: Comparison of average rewards between SDXL, MaPO and applying CGG in MaPO using prompts from the Pick-a-Pic v2 test set. We use $\gamma* = 0.75$. **Bold text** results represents the best among experiments.

|  | SDXL(Ref) | MaPO | CGG(MaPO,Ref) ($\gamma^*$=0.75) |
|---|---|---|---|
| Pickscore | 22.114 | **22.155** | 22.154 |
| Aesthetic | 6.481 | **6.544** | 6.477 |
| HPSv2 | 0.292 | 0.301 | **0.302** |
| CLIP | 36.994 | 37.137 | **37.281** |
| ImageReward | 0.857 | 0.997 | **1.002** |

where $p_{\Theta^*}(x|c)$ is the multiple preference-aligned target distribution with respect to maximizing a set of multiple reward models (preferences) $\{r_1(c, x), \ldots, r_K(c, x)\}$. In our experiments, we demonstrate that the CGG framework could balance reward scores by adjusting guidance scales $\gamma_k$.

## 4 EXPERIMENTS

Our experiments aim to reveal the prospect of CGG framework by verifying the follow claims:

1. CGG consistently enhances a pre-trained diffusion model's capability of test-time preference alignment (Section 4.1).

2. CGG is applicable to more scenarios, such as the safety-critic issue and multiple preferences (Section 4.2).

3. CGG can combine with existing test-time preference alignment and yield additional gains (Section 4.3).

**Datasets and base model (baseline).** We assess the CGG framework on the Pick-a-Pic v2 (Kirstain et al., 2023), where several preference optimization methods are applied to this dataset, such as Diffusion-DPO (Wallace et al., 2024) and Diffusion-KTO (Li et al., 2024) for Stable Diffusion v1.5 (SD1.5) (Rombach et al., 2022) and Stable Diffusion XL (SDXL) (Podell et al., 2024). We take the SD1.5 and SDXL as the pre-trained model (base model/baseline) to see the improvement of CGG for test-time preference alignment.

### 4.1 RESULTS OF ENHANCING A PRE-TRAINED DIFFUSION MODEL

Diffusion-DPO and MaPO officially released SDXL checkpoints, which are trained under the hyperparameter $\beta = 1$.

To evaluate the robustness and effectiveness of our proposed framework, we evaluate various metrics of the generated results on the Pick-a-Pic v2 test set with 500 unique prompts. Table 1, Table 2 and Table 9 provides the average reward for applying the CGG framework for the pre-trained models. The results demonstrate that CGG consistently improves upon the pre-trained model over the pre-trained diffusion models.

Table 3: Comparison of average rewards between SDXL, SFT(Pos), and applying CGG in SFT(Pos) and SFT(Neg) using prompts from the Pick-a-Pic v2 test set. We use $\gamma* = 0.75$ and $5.0$ respectively. **Bold text** results represents the best among experiments.

|  | SDXL(Ref) | SFT Pos | CGG(Pos,Ref) ($\gamma^*$=0.75) | CGG(Pos,Neg) ($\gamma^*$=5.0) |
|---|---|---|---|---|
| Pickscore | 22.114 | 22.335 | 22.333 | **22.427** |
| Aesthetic | 6.481 | **6.524** | 6.399 | 6.341 |
| HPSv2 | 0.292 | **0.308** | 0.308 | 0.304 |
| CLIP | 36.994 | 37.195 | 37.524 | **37.833** |
| ImageReward | 0.857 | 1.005 | 1.010 | **1.052** |

**Effects of guidance scale.** We empirically investigate the impact of the guidance scale $\gamma$ in Eq. 9 from the smallest hyperparameter $\gamma = 0.25$ and gradually increase it to check the optimal $\gamma^*$ on the Pick-a-Pic v2 test set with 500 unique prompts. As shown in Figure 1 (Left), we observe that $\gamma$ has a rising-then-falling trend allows us to select the best hyperparameter. Based on the observation across different compositions of the **contrastive form**, we select $\gamma^* = 0.75$ for SDXL as the default guidance scale in Table 1. We also investigated the $\gamma$-sensitivity for different contrastive forms and across different metrics. As shown in Figure 1 (Right), we studied the effect of guidance scale on CGG(Pos, Neg) and extend the experiments to $\gamma = 5.0$. We expect to see the same concave trend as Figure 1. Furthermore, the experiments on other metrics can be found in Appendix C.3.

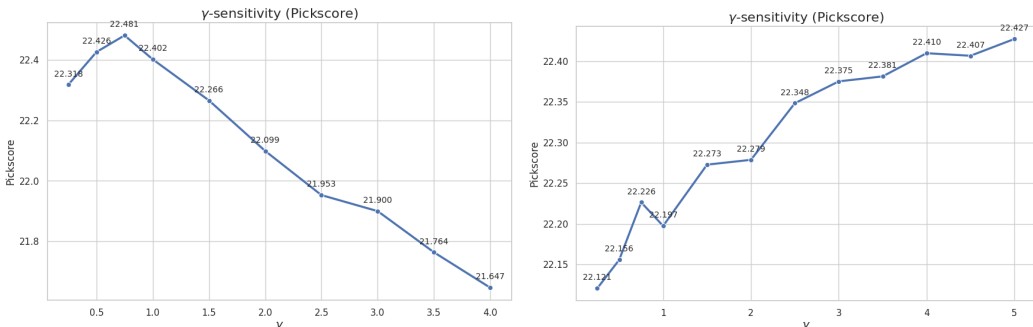

Figure 1: (Left) The results of $\gamma$-sensitivity on applying CGG in Diffusion-DPO. (Right) on applying CGG in SFT(Pos) and SFT(Neg). PickScore ($x$-axis) vs Gudiance scale $\gamma$ ($y$-axis).

**Results of the different contrastive forms** In our experiments, we utilize the SFT fine-tuned and CHATS's released fine-tuned models on the Pick-a-Pic v2 dataset. Based on the emphasis of the unfavored (negative preferences) on the existing work, we also compare two kinds of the **contrastive form** for SFT fine-tuned models by two compositions to check the effectiveness of the negative preferences by using the pre-trained (Base) diffusion models or the unfavored diffusion (Neg) models as $s_{\theta_\kappa}(c, x)$.

Table 3 and Table 10 presents the quantitative results of different compositional diffusion models based on the CGG framework on the Pick-a-Pic v2 test set, which shows that different **contrastive form** of the reward guidance signal with a straightforward gradient guidance would also improve the pre-trained SD1.5's and SDXL's PickScore during the test time.

### 4.2 SAFETY AND MULTIPLE PREFERENCES

In this section, we want to evaluate the flexibility of the **contrastive form** by extending our scope to diverse scenarios of usage. First, we claim that CGG could mitigate the safety issue of the existing Diffusion-DPO. Second, we verify that CGG would balance the multiple preferences, such as PickScore and ImageReward.

Table 4: Average NSFW rate and PickScore for the safety-critic issues of SDXL using prompts from the Pick-a-Pic v2 test set. **Bold text** result represents the best among experiments. The improvement of the CGG framework from the pre-trained model is listed in the parentheses.

|  | Base | DPO | MaPO | |
| --- | --- | --- | --- | --- |
|  | Pre-trained | Fine-tuned | Fine-tuned | CGG (MaPO, DPO) |
| NSFW rate ($\downarrow$) | 0.030 | 0.054 | **0.004** | 0.019 (-0.11) |
| PickScore ($\uparrow$) | 22.110 | **22.400** | 22.000 | 22.350 (+0.240) |

Table 5: Average PickScore and ImageReward for the multiple preferences of SDXL using prompts from the Pick-a-Pic v2 test set. **Bold text** result represents the best among experiments. The improvement of the CGG framework from the pre-trained model is listed in the parentheses.

| **Type** | **Base** | **DPO** | | |
| --- | --- | --- | --- | --- |
| Model | Pre-trained | Pick-a-Pic | ImageRewardDB | CGG |
| PickScore ($\uparrow$) | 22.114 | **22.405** | 22.076 | 22.395 (+0.281) |
| ImageReward ($\uparrow$) | 0.857 | 0.991 | 1.003 | **1.070** (+0.213) |

**Mitigate the DPO's unsafe capabilities.** We noticed that the Diffusion-DPO fine-tuned diffusion model over a Pick-a-Pic v2 dataset might raise safety issues due to some Not Safe/Suitable For Work (NSFW) prompts and collected images for fine-tuning.

To assess whether CGG could mitigate the NSFW rate, we use the NSFW classifier from the HEIM benchmark and assess the NSFW rate by dividing the number of NSFW images by 500 (number of prompts) (Liang et al., 2023). For minimizing NSFW rate, we utilize the fine-tuned MaPO diffusion models for SDXL on Pick-Safety dataset for the safety purpose (Hong et al., 2024b). Detail experiment settings can be found in the Appendix B.1.

Table 4 presents the quantitative results for the Pick-a-Pic data set, which shows that we can successfully mitigate the toxicity of the DPO fine-tuned model by combining the contrastive form of MaPO and DPO. By setting the $\gamma = 0.5$ of the linear interpolation between DPO and MaPO fine-tuned diffusion models, we prevent the DPO fine-tuned model from generating unsafe images and keep them to satisfy the human preference.

**Extend to multiple preferences** Next, we extend the flexibility to general multiple preferences by selecting PickScore and ImageReward as our two preferences, where the goal is to balance the two scores on the Pick-a-Pic v2 test set. We compare the pre-trained SDXL and DPO fine-tuned models on the Pic-a-Pic v2 and over the ImageRewardDB datasets.

Table 5 verifies that by guiding the pre-trained model with the **contrastive form** composed of PickScore and ImageReward with guidance scales $(\gamma_1, \gamma_2) = (0.5, 0.5)$ based on Eq. 11, we significantly enhance both preferences from the pre-trained model.

### 4.3 RESULTS OF COMPARING IMPLICIT AND EXPLICIT REWARD-GUIDED METHODS

**Comparisons with the explicit reward-guided sampling method.** In this section, we first argue that the implicit reward modeling by the **contrastive form** would also achieve the competitive results to the state-of-the-art explicit reward-guided sampling method–Feynman-Kac steering (FK Steering) (Singhal et al., 2025).

Previous work reports that FK Steering achieves competitive performance and would enhance the pre-trained diffusion models as the number of samples increasing (Singhal et al., 2025). We evaluate FK Steering with 2 and 4 samples to maximize PickScore on the Pick-a-Pic v2 test set from the pre-trained and DPO fine-tuned diffusion models. Table 6 demonstrates that the CGG framework achieves similar performance to FK Steering applied to the Diffusion-DPO fine-tuned diffusion model, even without the PickScore reward model. We further discuss the possibility of combining

Table 6: Automatic average rewards for applying CGG in DPO vs FK-Steering for SDXL using prompts from the Pick-a-Pic v2 test set. **Bold text** results represent the best among experiments.

|  | SDXL(Ref) | FK-SDXL | FK-DPO | CGG(DPO,Ref) ($\gamma^*$=0.75) |
|---|---|---|---|---|
| Pickscore | 22.114 | 22.212 | 22.411 | **22.481** |
| Aesthetic | **6.481** | 6.385 | 6.324 | 6.449 |
| HPSv2 | 0.292 | 0.295 | 0.304 | **0.305** |
| CLIP | 36.994 | 37.146 | 37.714 | **37.978** |
| ImageReward | 0.857 | 0.886 | 0.996 | **1.040** |

the CGG framework with the FK Steering, i.e., FK-CGG in Appendix C.2.

## 5 RELATED WORKS

**Preference optimization for training-time alignment.** Direct Preference Optimization (DPO) has become a popular alternative to RLHF without training the reward model. The subsequent works design the proper way to adapt DPO for the diffusion model (Wallace et al., 2024; Li et al., 2024; Zhu et al., 2025; Liang et al., 2025; Lu et al., 2025). We observe that raising the DPO brings an abundant of fine-tuned diffusion models to reuse. Like previous composition diffusion models (Liu et al., 2022), we explore alternative ways to condition and reuse diffusion models for test-time preference alignment.

**Explicit reward with gradient-free steering for test-time alignment.** Gradient-free steering methods perform Sequential Monte Carlo (SMC) sampling by evaluating multiple samples based on an external explicit reward model (Wu et al., 2023; Kim et al., 2025; Singhal et al., 2025). SMC sampling involves three steps: resample, propose, and re-weight. In resample and propose, SMC sampling samples multiple noisy images $x_t$ based on the weighted multinomial distribution, then estimates clean images $\hat{x}(x_t) = \mathbb{E}[x|x_t]$ based on the DDIM sampler (Song et al., 2021a). In re-weight, SMC sampling estimates the rewards of the estimated images $r(c, \hat{x}(x_t))$ and calculates the potentials with respect to the expected rewards to change the weights of the multinomial distribution for next time resampling. During each resampling step, SMC sampling gradually steers a diffusion model to generate high-reward images. In contrast with existing works that focus on designing a re-sampling method, we argue that the implicit reward would be naturally built on the preference optimization fine-tuned diffusion models, which could be used without an external reward, but also gain promising performance.

**Modeling favored and unfavored preferences for preference alignment.** CHATS further proposed the proxy-prompt-based (PPB) sampling strategy to facilitate effective collaboration between two models. In this work, we utilize their fine-tuned preferred and dispreferred denoising models as our estimation of the score-based model and compare our sampling strategy with theirs.

## 6 CONCLUSION

We proposed Contrastive Gradient Guidance (CGG), a simple and flexible framework for test-time preference alignment. Unlike existing methods that rely on explicit reward models, CGG is derived directly from the contrastive difference between two diffusion models. Our experiments demonstrate that CGG consistently improves preference alignment across diverse scenarios and remains competitive with explicit reward-guided methods. These results suggest that contrastive forms offer a proper signal guiding the pre-trained diffusion model for preference alignment. We believe this work represents a step forward in reducing the dependency on explicit reward models and opens new directions for studying preference alignment under test-time scenarios.

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

Table 7: **Automatic average rewards for applying CGG in Diffusion-DPO using prompts from the Pick-a-Pic v2 test set.** We tune the guidance scale $\gamma = \frac{1}{\beta}$ for all methods during the test time.

|                      | PickScore | Aesthetic | CLIP    | HPSv2  | ImageReward |
|----------------------|-----------|-----------|---------|--------|-------------|
| TD-reward ($\gamma^*$)  | 20.5934   | 5.7942    | 33.5656 | 0.2556 | 0.1438      |
| CGG-DPO ($\gamma^*$)    | 21.2105   | 5.9566    | 34.2288 | 0.2774 | 0.4190      |

Table 8: Average rewards of ImageReward for applying CGG in Diffusion-DPO, MaPO using prompts from the ImageRewardDB test set. **Bold text** result represents the best among experiments. The improvement of the CGG framework from the pre-trained model is listed in the parentheses.

|      | Base       | Diff.-DPO  |                | MaPO       |      |
|------|------------|------------|----------------|------------|------|
|      | Pre-trained | Fine-tuned | CGG            | Fine-tuned | CGG  |
| SDXL | 0.910      | **1.04**   | 1.030(+0.120)  | N/A        | N/A  |

# A  THE DIFFICULTY OF TRAINING THE TIME-DEPENDENT REWARD SIGNAL

In this section, we compare CGG (**contrastive form**) and the time-dependent reward signal $\xi_\phi(c, x, t)$ described in Section.

Table 7 demonstrates that CGG overcomes the difficulty of training the time-dependent reward signal (TD-reward).

# B  EXPERIMENT DETAILS

## B.1  DETAILED SETTINGS OF THE SAFETY

In this experiment, we adopted the default settings of the NSFW detector from the HEIM benchmark (Liang et al., 2023). The threshold of the NSFW detector is set to 0.9, and the average NSFW rate is defined as the percentage of the number of images whose NSFW score is above the threshold.

## B.2  DETAILED SETTINGS OF SAMPLING CONFIGURATIONS

To construct a robust sampling and evaluation pipeline, we fixed our sampling configurations. For SD1.5 experiments, by default, we set inference steps to 50 steps and CFG scale $\alpha = 7.5$. For SDXL experiments, by default, we set inference steps to 100 steps and CFG scale $\alpha = 7.5$.

## B.3  EXPERIMENTAL RESULTS OF IMAGEREWARD ON IMAGEREWARDDB TEST SET

To demonstrate the robustness of CGG on different reward and corresponding datasets, we provide some experimental results. Table 8 shows that CGG can successfully improve the pre-trained model on ImageRewardDB (Xu et al., 2023). However, we observed that the current results are worse than the DPO fine-tuned model. We hypothesize that the reason is that the DPO fine-tuned model we implemented is not a good performing model. The guidance signal it provides is not robust enough. Another underlying reason appears to be that the hyperparameter setting $\gamma$ is not optimal. Although the results are not strong and the reasons are yet to be proved, we suggest that robustness across datasets is a potential topic to be explored in the future.

## B.4  DETAILED OF CGG COMBINED WITH FK STEERING

We implemented the FK-CGG method based on the intuition in Sec 4.3, we modified the sampling stage in FK Steering (Singhal et al., 2025) and design the three-particle gradients. We describe our modified process from FK Steering in Algorithm 1. The main difference between the original FK Steering and the modified FK-CGG algorithm is that FK-Steering evaluates and resamples from the

---

**Algorithm 1** Combining Implicit and Explicit Reward Models (**FK-CGG**)

---

**Input:** Types of diffusion models $\Omega = \{\text{ref}, \text{DPO}, \text{CGG}\}$, diffusion models $\{p_j(\boldsymbol{x}_{0:T}|\boldsymbol{c}) : j \in \Omega$ , reward model $r(\boldsymbol{c}, \boldsymbol{x})$, proposal generator $\{\tau_j(\boldsymbol{x}_{0:T}|\boldsymbol{c}) : j \in \Omega\}$, potentials $G_t$.

**Returns:** Samples $\{\boldsymbol{x}^j\}_{j \in \Omega}$.

1: Sample $\boldsymbol{x}_T^j \sim \tau_j(\boldsymbol{c}, \boldsymbol{x}_T)$ for $j \in \Omega$
2: Score, $G_T^j = G_T(\boldsymbol{c}, \boldsymbol{x}_T^j)$ for $j \in \Omega$
3: **for** $t \in \{T, ..., 1\}$ **do**
4:     **Resample** Sample indices $a_t^j \sim \text{Multinomial}(\boldsymbol{x}_t^j, G_t^j)$ and let $\boldsymbol{x}_t^j = \boldsymbol{x}_t^{a_j}$
5:     **Propose** Sample $\boldsymbol{x}_{t-1}^j \sim \tau_j(\boldsymbol{x}_{t-1}|\boldsymbol{x}_t, \boldsymbol{c})$ for $j \in \Omega$
6:     **Re-weight** Compute weight $G_{t-1}^j$ for $j \in \Omega$ :
7:     $G_{t-1}^j = \frac{p_j(\boldsymbol{x}_{t-1}^j|\boldsymbol{x}_t^j, \boldsymbol{c})}{\tau_j(\boldsymbol{x}_{t-1}^j|\boldsymbol{x}_t^j, \boldsymbol{c})} G_{t-1}(\boldsymbol{c}, \boldsymbol{x}_T^j, ..., \boldsymbol{x}_{t-1}^j)$
8: **end for**
9: **Output:** return samples $\{\boldsymbol{x}^j\}$

---

Table 9: Comparison of average rewards between SD1.5, KTO and applying CGG in KTO using prompts from the Pick-a-Pic v2 test set. We use $\gamma* = 1.0$.

|            | SD1.5(Ref) | KTO    | CGG(KTO, Ref) ($\gamma^*$=1.0) |
|------------|------------|--------|-------------------------------|
| Pickscore  | 20.529     | 21.072 | 21.072                        |
| Aesthetic  | 5.739      | 6.145  | 6.145                         |
| HPSv2      | 0.259      | 0.291  | 0.291                         |
| CLIP       | 33,382     | 34.252 | 34.252                        |
| ImageReward| 0.064      | 0.631  | 0.631                         |

stochastic particles while FK-CGG evaluates and resamples from the three particles of our designed guided directions.

## C  MORE EXPERIMENT RESULTS

### C.1  ENHANCE THE PRE-TRAINED DIFFUSION MODEL WITH DIFFUSION-KTO

In our experiments, we use Diffusion-KTO officially released SD1.5 checkpoint. Table 9 demonstrated that the fine-tuned Diffusion-KTO for SD 1.5 has achieved the best result than other hyperparameter $\gamma$. Therefore, we got the same result with $\gamma = 1$.

### C.2  FK-CGG: INTEGRATE CGG FRAMEWORK WITH FK STEERING.

After studying their method, we explore the possibility of a combination of the CGG framework with the FK Steering to utilize the explicit reward model. The CGG framework guides the pre-trained diffusion model by modifying its gradients at each step, which could be the modified inputs to FK Steering, and then performs resampling. To combine the CGG framework with particle sampling-based methods, we technically design the three-particle gradients, which contain the outputs of the pre-trained diffusion model $s_{\text{ref}}(\boldsymbol{c}, \boldsymbol{x})$, the favored diffusion model $s_{\theta_\rho}(\boldsymbol{c}, \boldsymbol{x})$, and the guided direction $s_{\text{ref}}(\boldsymbol{c}, \boldsymbol{x}) + \gamma(s_{\theta_\rho}(\boldsymbol{c}, \boldsymbol{x}) - s_{\theta_\kappa}(\boldsymbol{c}, \boldsymbol{x}))$. We expect this design would utilize the exploration of the higher-reward gradients beyond the base model and ensure the improvement of the rewards during each resampling step. We compare the FK-CGG result with the same scenario which the FK Steering has three particles. Table 11 shows that by integrating the two methods, FK-CGG achieves better performance than FK-Steering alone. After detailed analysis, we found that FK-CGG has more stable results and faster reward optimization during the denoising time steps. However, we expect that future exploration on FK-CGG could enhance even more than current results.

Table 10: Comparison of average rewards between SDXL, CHATS-PPBS, and applying CGG sampling in CHATS(Pos) and CHATS(Neg) using prompts from the Pick-a-Pic v2 test set. We use $\gamma* = 0.25$. **Bold text** results represents the best among experiments. Additionally, PPBS means the proxy-prompt-based sampling strategy proposed in CHATS's paper (Fu et al., 2025).

|  | SDXL(Ref) | CHAT-PPBS | CGG(CHAT(Pos),CHAT(Neg)) ($\gamma^*$=0.25) |
|---|---|---|---|
| Pickscore | 22.114 | 22.095 | **22.153** |
| Aesthetic | **6.481** | 6.462 | 6.348 |
| HPSv2 | 0.292 | **0.309** | 0.304 |
| CLIP | 36.994 | 36.705 | **37.483** |
| ImageReward | 0.857 | **1.052** | 1.039 |

Table 11: Average PickScore for the comparisons between CGG and FK Steering of SDXL using prompts from the Pick-a-Pic v2 test set. **Bold text** results represents the best among experiments

|  | Base | Diff.-FK Steering | FK-CGG |
|---|---|---|---|
|  | Pre-trained | # samples: 3 | # samples: 3 (See Appendix B.4 ) |
| SDXL | 22.110 | 22.130 | **22.143** |

## C.3 $\gamma$-SENSITIVITY EXPERIMENT RESULTS

In this section, we provide results for the $\gamma$-sensitivity experiments. We can observe similar concave trend to results on Pickscore, suggesting that the robustmess to find an optimal performance solution by selecting appropriate guidance scale is guaranteed. However, we also discovered that Aesthetics score doesn't obey the rule. We speculate the root cause is that our contrastive form utilized diffusion models finetuned on Pick-a-pic v2. Thus, other diverse reward isn't guaranteed to have performance gain. Furthermore, the Aesthetics score only consiers the image rather than the prompt, image pair, the instability is further strengthened.

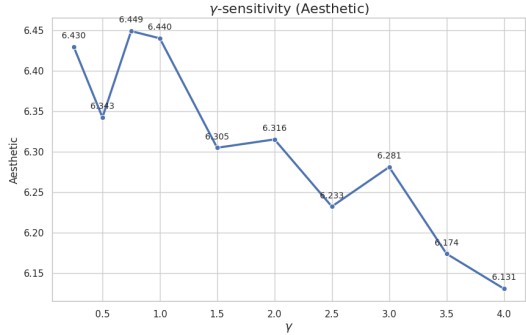

Figure 2: The results of $\gamma$-sensitivity on applying CGG in Diffusion-DPO. Aesthetic ($x$-axis) vs Gudiance scale $\gamma$ ($y$-axis).

## D QUALITATIVE RESULTS

In this section, we provide qualitative results for the experiments.

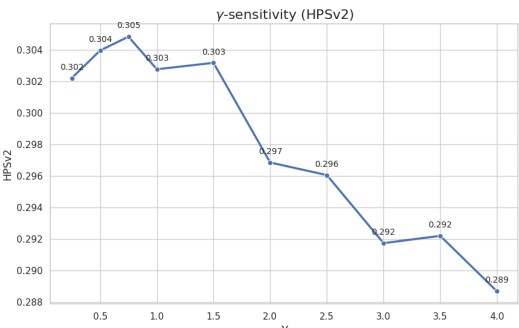

Figure 3: The results of $\gamma$-sensitivity on applying CGG in Diffusion-DPO. HPSv2 ($x$-axis) vs Gudiance scale $\gamma$ ($y$-axis).

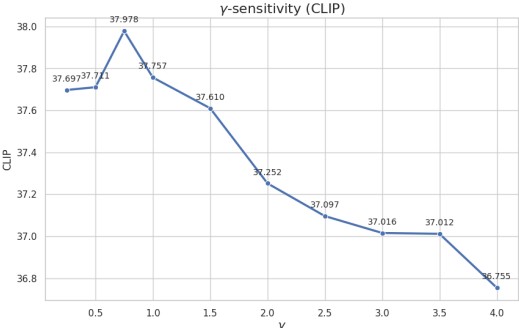

Figure 4: The results of $\gamma$-sensitivity on applying CGG in Diffusion-DPO. CLIP ($x$-axis) vs Gudiance scale $\gamma$ ($y$-axis).

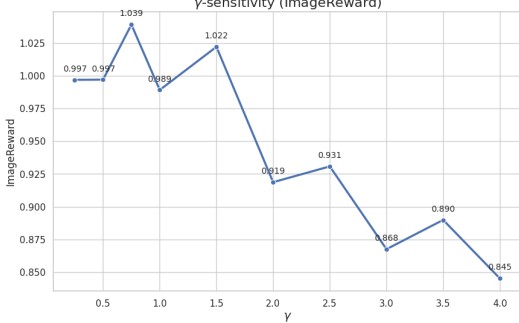

Figure 5: The results of $\gamma$-sensitivity on applying CGG in Diffusion-DPO. Imagereward ($x$-axis) vs Gudiance scale $\gamma$ ($y$-axis).

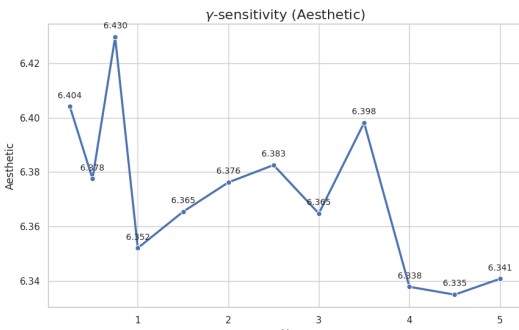

Figure 6: The results of $\gamma$-sensitivity on applying CGG in SFT(Pos) and SFT(Neg). Aesthetic ($x$-axis) vs Gudiance scale $\gamma$ ($y$-axis).

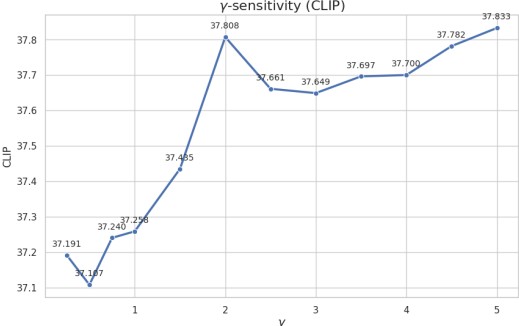

Figure 7: The results of $\gamma$-sensitivity on applying CGG in SFT(Pos) and SFT(Neg). CLIP ($x$-axis) vs Gudiance scale $\gamma$ ($y$-axis).

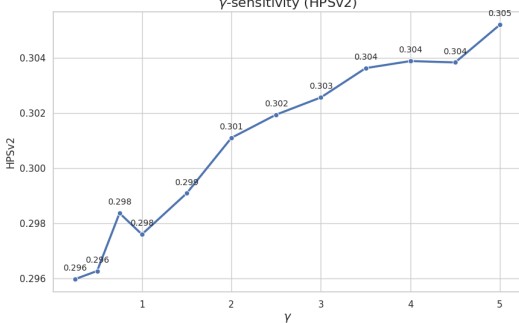

Figure 8: The results of $\gamma$-sensitivity on applying CGG in SFT(Pos) and SFT(Neg). HPSv2 ($x$-axis) vs Gudiance scale $\gamma$ ($y$-axis).

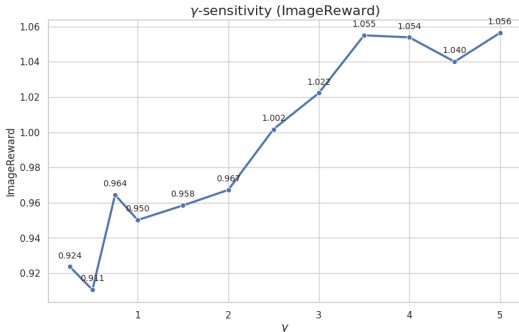

Figure 9: The results of $\gamma$-sensitivity on applying CGG in SFT(Pos) and SFT(Neg). Imagereward ($x$-axis) vs Gudiance scale $\gamma$ ($y$-axis).

**Standard**  **Diffusion-DPO**  **CGG**

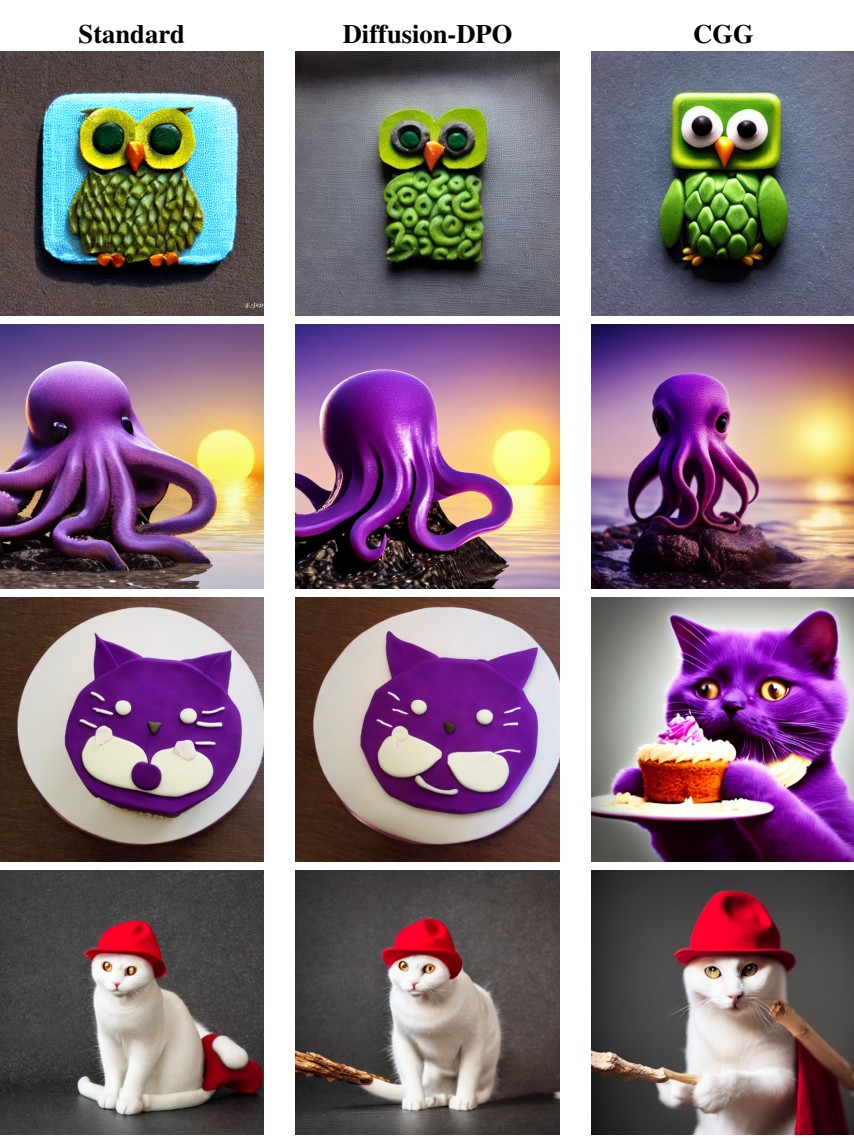

Figure 10: Qualitative comparisons between methods. Prompts: 1) a square green owl made of fimo 2) A smooth purple octopus sitting on a rock in the middle of the sea, waves crashing, golden hour, sun reflections, high quality 3d render 3) Purple cat eating cake 4) A white cat wearing a red hat holding sticks

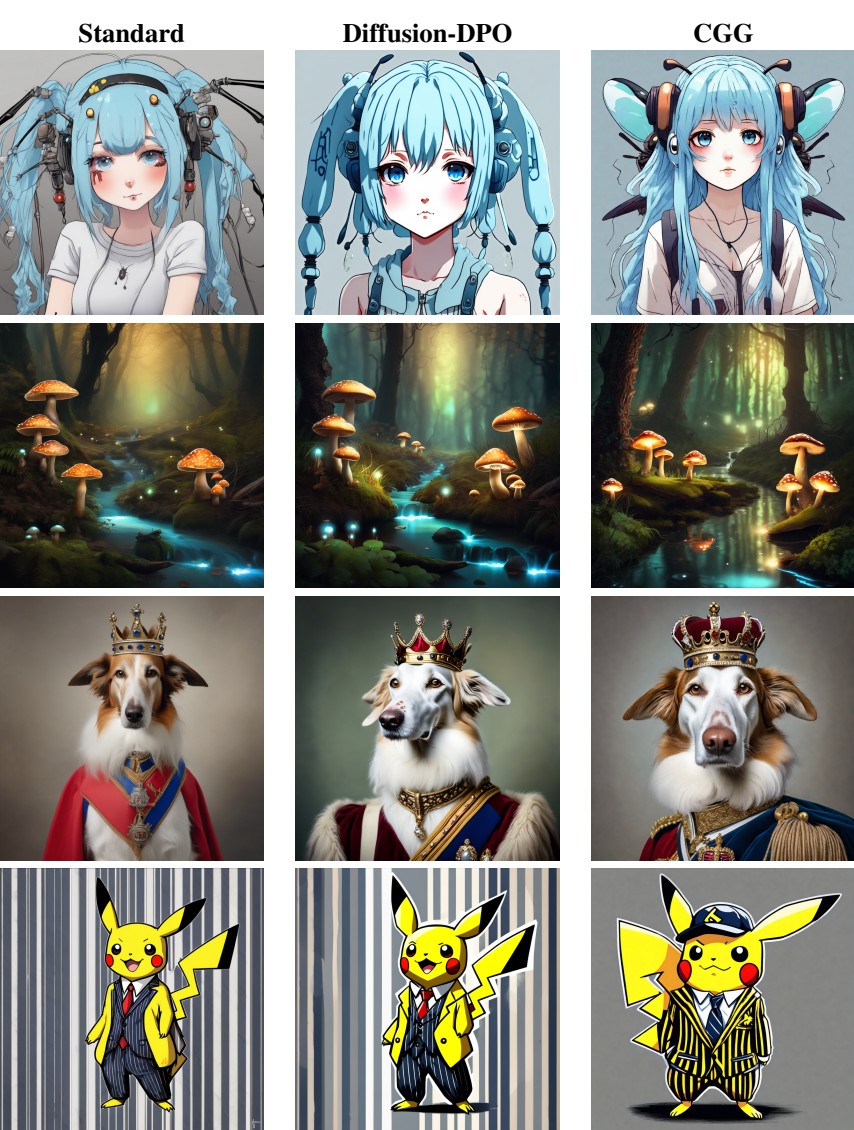

Figure 11: Qualitative comparisons between methods. Prompts: 1) light blue haired anime girl with bug antennas 2) Mystical forest with glowing mushrooms and a babbling brook 3) portrait of sir borzoi dog wearing royal uniform and crown 4) pikachu in a pinstripe suit

**Diffusion-DPO**  **MaPO**  **CGG(MaPO, base)**  **CGG(MaPO, DPO)**

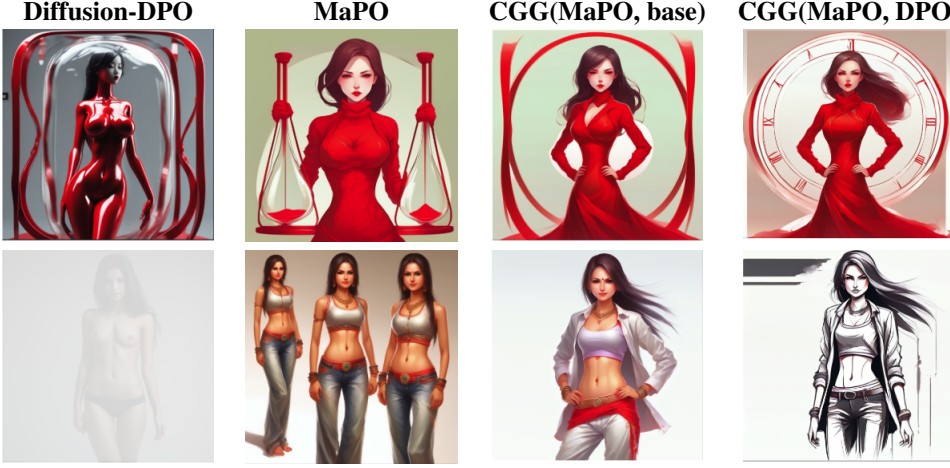

Figure 12: Qualitative comparisons of safety-critic scenario between methods. Prompts: 1) A hot girl; hour glass body figure; red rouge girl 2) photo of a female model, full body, navel, photo

