# OpenReview forum: "Contrastive Gradient Guidance for Test-time Preference Alignment of Diffusion Models"
_ICLR.cc/2026/Conference — Submitted to ICLR 2026_

### Official Review · Reviewer_cSmV · 2025-10-30

**Soundness:** 3
**Presentation:** 3
**Contribution:** 2
**Rating:** 4
**Confidence:** 4

**Summary:**

This paper proposes Contrastive Gradient Guidance (CGG), a method that use fine-tuned policy models to guide the sampling process at test time. The authors conducted experiments on image quality, safety, and multiple preference setup, demonstrating the effectiveness of the proposed method.

**Strengths:**

1. The proposed method is highly flexible and can be plugged in many different base models, finetuned models (DPO, KTO), and even training-free steering (FK-steering).
2. The proposed method work for a variety of application scenarios such as general preference (PickScore), safety, and combining multiple rewards without training new models.

**Weaknesses:**

1. The paper has poor presentation. In particular, many tables are hard to read.  For example, Table 7 does not have any midrules or booktabs. It is unclear what "*" means in Table 1, as it shows up on all CGG numbers. Table 4 and Table 3 has inconsistent header format despite showing results of similar structure. I strongly encourage the author to revise the presentation of results for the final version.
2. The paper is not well-motivated. I understand the need to get rid of "explicit reward models" for test-time alignment methods, however this paper proposes instead we need to have a set of finetuned policy models (e.g. DPO,KTO), which seems to be more difficult to obtain. The author should clearly discuss on what kind of usecases is this assumption justified. For example, I can imagine if we have a newly released foundational model, methods that rely on reward models can still work, but CGG cannot work without having someone else releasing a fine-tuned model first. It is unclear why the setup of CGG is more preferable than works using explicit reward models.

**Questions:**

In Table 1, KTO and CGG has exactly the same number 21.072, can the author confirm this? I understand when $\gamma$ is set to 1, KTO and CGG should have exactly the same results. However the author used $\gamma=3.5$, why are the number exactly the same?

---

> ### Author Response · Authors · 2025-12-04
>
> > 1. The paper has poor presentation. In particular, many tables are hard to read. For example, Table 7 does not have any midrules or booktabs. It is unclear what "*" means in Table 1, as it shows up on all CGG numbers. Table 4 and Table 3 has inconsistent header format despite showing results of similar structure. I strongly encourage the author to revise the presentation of results for the final version.
>
> Thank you for your suggestions on enhancing the table presentation. We have updated all Tables and only highlighted the best results for the different models and settings. We also added the improvement of the CGG to emphasize the effectiveness of our method for enhancing the pre-trained diffusion models.
>
> > 2. The paper is not well-motivated. I understand the need to get rid of "explicit reward models" for test-time alignment methods, however this paper proposes instead we need to have a set of finetuned policy models (e.g. DPO,KTO), which seems to be more difficult to obtain. The author should clearly discuss on what kind of usecases is this assumption justified. For example, I can imagine if we have a newly released foundational model, methods that rely on reward models can still work, but CGG cannot work without having someone else releasing a fine-tuned model first. It is unclear why the setup of CGG is more preferable than works using explicit reward models.
>
> As we mentioned in the General Response, we want to demonstrate the possibility of the compositional distributions could replace the (relative large) reward model, which stands for the idea with this position paper (1): *compositional generative approach enables us to learn distributions in a more data-efficient manner*.
>
> Besides, we also want to recall that the implicit reward model is a critical idea from the DPO (2), which aims to skip the complicated process of training a reward model and fine-tune the pre-trained generative model from it.
>
> > 3. In Table 1, KTO and CGG have exactly the same number, 21.072. Can the author confirm this? I understand when is set to 1, KTO and CGG should have exactly the same results. However the author used  , why are the number exactly the same?
>
> After we double-check the experimental results, we notice that the fine-tuned Diffusion-KTO for SD1.5 has achieved the best result than other hyper-parameter $\gamma$. Therefore, we got the same result with $\gamma = 1$.

---

### Official Review · Reviewer_i9dk · 2025-10-30

**Soundness:** 3
**Presentation:** 3
**Contribution:** 3
**Rating:** 6
**Confidence:** 3

**Summary:**

This paper introduces Contrastive Gradient Guidance (CGG), a test-time framework for aligning diffusion model outputs with human preferences without relying on explicit reward models. CGG leverages the gradient of a log-likelihood ratio between favored and unfavored diffusion model distributions to produce an implicit reward guidance signal. This signal adjusts the model sampling path in real time to produce more preference-consistent outputs. The authors present a theoretical connection between CGG and reward-guided diffusion methods, as well as the established classifier-free guidance (CFG) mechanism. Experiments on Pick-a-Pic v2 and related datasets show that CGG improves preference alignment metrics such as PickScore and ImageReward, demonstrating flexibility across safety-critical and multi-preference scenarios.

**Strengths:**

The paper presents a novel and conceptually elegant approach to test-time alignment of diffusion models. Its most significant contribution lies in eliminating the reliance on explicit reward models, which are costly to train and often difficult to adapt to new domains or user preferences. By framing alignment as a contrastive gradient problem, the authors provide a computationally efficient solution that maintains flexibility and broad applicability.

**Weaknesses:**

1) While the proposed idea is promising, the empirical evidence supporting CGG’s effectiveness should be strengthened.

2) The performance gains reported in Tables 1 and 2 are relatively modest, and in some cases, CGG does not achieve state-of-the-art scores. The authors should provide deeper justification or analysis to explain the observed improvements and limitations.

3) Furthermore, the evaluation relies solely on automated preference metrics, which may not fully capture human judgment. A user study or human evaluation comparing CGG-generated results with baseline methods may provide better results and would greatly strength the paper’s claim.

4) Finally, the empirical validation currently focuses only on Stable Diffusion 1.5 and SDXL; extending the experiments to additional models would improve the generality and credibility of the findings.

**Questions:**

Please provide more experimental results and justification in the rebuttal period.

---

> ### Author Response · Authors · 2025-12-04
>
> > 1. While the proposed idea is promising, the empirical evidence supporting CGG's effectiveness should be strengthened.
>
> We added more evaluation results and the scope of the hyperparameters $\gamma$ for the CGG to enrich the supportive empirical evidence of CGG's effectiveness.
>
> In the next version, we plan to add more experiments of utilizing the CGG for the Stable Diffusion 3.5 and FLUX models, which are Diffusion Transformers (DiTs)-based models.
>
> > 2. The performance gains reported in Tables 1 and 2 are relatively modest, and in some cases, CGG does not achieve state-of-the-art scores. The authors should provide deeper justification or analysis to explain the observed improvements and limitations.
>
> Thank you for pointing out the concern of the performance gains reported in Table 1 and Table 3 (which replaced the original Table 2).
> We aim to clarify the messages conveyed in those tables, extending beyond the focus on SOTA performance.
> First, Table 1 demonstrates that the CGG(DPO, Ref) can guide the pre-trained diffusion model with the DPO-fine-tuned diffusion models, thereby enhancing both the pre-trained and DPO-fine-tuned models.
> Second, Table 3 (originally Table 2) highlights not only that PO-based models can guide the pre-trained diffusion model, but also that we can use supervised fine-tuning (SFT) for the two small subsets (images with positive feedback and negative feedback) to guide the pre-trained diffusion model. While this approach does not achieve the SOTA results, we affirm the possible implementations to utilize the CGG framework.
>
> > 3. Furthermore, the evaluation relies solely on automated preference metrics, which may not fully capture human judgment. A user study or human evaluation comparing CGG-generated results with baseline methods may provide better results and would greatly strength the paper's claim.
>
> Thanks for your suggestions. We will add human evaluation results in the next version.
>
> > 4. Finally, the empirical validation currently focuses only on Stable Diffusion 1.5 and SDXL; extending the experiments to additional models would improve the generality and credibility of the findings.
> > - Please provide more experimental results and justification in the rebuttal period.
>
> In the next version, we plan to add more experiments of utilizing the CGG for the Stable Diffusion 3.5 and FLUX models, which are Diffusion Transformers (DiTs)-based models.

---

### Official Review · Reviewer_Dq7k · 2025-10-31

**Soundness:** 2
**Presentation:** 3
**Contribution:** 2
**Rating:** 2
**Confidence:** 4

**Summary:**

This paper proposes Contrastive Gradient Guidance (CGG), a test-time alignment method for diffusion models that avoids explicit reward modeling.
CGG derives a guidance signal from the contrastive difference between favored and unfavored distributions, expressed as the gradient of their log-likelihood ratio.
By steering a pretrained diffusion model using this signal, CGG enables preference-aligned generation without additional training.
Experiments on Pick-a-Pic v2 demonstrate improved PickScore over Diffusion-DPO/KTO baselines, while the framework also adapts to safety-critical and multi-preference settings and can be combined with explicit reward-guided methods such as FK-Steering.

**Strengths:**

1. This paper proposes a simple yet practical inference-time preference alignment method that draws an analogy to classifier guidance (CG).
2. It provides a practical approach that can reuse existing pretrained weights and preference-tuned weights.
3. In addition to standard preference alignment, the method also improves performance across multiple tasks such as NSFW rate reduction and multi-preference alignment.

**Weaknesses:**

1.The novelty of the proposed method is uncertain. The definitions of ρ (line 181) and κ (line 183) are unclear, and the paper does not provide any numerical or theoretical analysis on how closely the formulation in Equation (7) approximates an actual reward function.

2.The guidance scale γ must be manually tuned, and as shown in Figure 1, the performance for SDXL even degrades as γ increases, making its effectiveness unclear.

3.The evaluation metrics are somewhat limited. It would be important to also include results on Aesthetics, CLIP, and HPS v2 to provide a more comprehensive evaluation.

**Questions:**

1. Could you provide a clearer explanation of the paragraph from lines 180 to 185?
   In addition, if you have any numerical or theoretical analysis on how closely the proposed formulation approximates the reward function, please share it.

2. Could you also share the evaluation results with Aesthetic, CLIP, and HPS v2, as well as the γ-sensitivity plot similar to Figure 1?

---

> ### Author Response · Authors · 2025-12-04
>
> > 1. The novelty of the proposed method is uncertain.
>
> The novelty of the CGG lies in viewing the reward model as the contrastive compositional favored and unfavored distributions, which enables us to guide the pre-trained diffusion model to generate a preference-aligned target distribution.
>
> > 1. The definitions of ρ (line 181) and κ (line 183) are unclear and the paper does not provide any numerical or theoretical analysis on how closely the formulation in Equation (7) approximates an actual reward function.
> > - Could you provide a clearer explanation of the paragraph from lines 180 to 185? In addition, if you have any numerical or theoretical analysis on how closely the proposed formulation approximates the reward function, please share it.
>
> Thanks for your comments. We updated the presentation of $\rho$ and $\kappa$ to reveal that Eq. (7) is our assumption for modeling the reward model with the contrastive form.
> Because we cannot directly estimate the $\log \frac{\rho(x \vert c)}{\kappa(x \vert c)}$ from the diffusion models, we propose an alternative way to verify Eq. (7).
> We plan to start with the positive and negative approach mentioned in Section 3 **Fine-tuned two diffusion models from pairwise preference data.**
> 1. Estimate the likelihood of two distributions based on the binary classifier on the pairwise preference dataset, which presents $\rho(x \vert c)$ and $\kappa(x \vert c) = 1 - \rho(x \vert c)$.
> 2. Estimate the $\log \frac{\rho(x \vert c)}{1 - \rho(x \vert c)}$.
> 3. Compare the relationship between $r(c, x)$ and $\log \frac{\rho(x \vert c)}{1 - \rho(x \vert c)}$.
>
> > 2. The guidance scale γ must be manually tuned, and as shown in Figure 1, the performance for SDXL even degrades as γ increases, making its effectiveness unclear.
>
> In our analysis of PO-based CGG in Figure 1, we discover that a sweet point exists between $[0, 1]$, which indicates the interpolation between the pre-trained model and the PO-based diffusion model yields the best result.
>
> In Figure 2, we further examine the SFT(Pos) and SFT(Neg) that continuously increase in performance as $\gamma$ is enlarged, indicating that we have not yet achieved the optimal hyperparameters for this case.
>
> Based on the current observations, while $\gamma$ needs to be manually tuned, we can determine the optimal value to select it automatically using a small validation set.
>
> > 3. The evaluation metrics are somewhat limited. It would be important to also include results on Aesthetics, CLIP, and HPS v2 to provide a more comprehensive evaluation.
> > - Could you also share the evaluation results with Aesthetic, CLIP, and HPS v2, as well as the γ-sensitivity plot similar to Figure 1?
>
> Thanks for your suggestions. We added Aesthetic, CLIP, HPS v2, and ImageReward in Tables 1 and 2, as well as the $\gamma$-sensitivity plot in Appendix C.

---

### Official Review · Reviewer_vy1u · 2025-11-01

**Soundness:** 2
**Presentation:** 3
**Contribution:** 2
**Rating:** 4
**Confidence:** 4

**Summary:**

Pre-trained diffusion models achieve impressive text-to-image generation quality, but aligning them with human preferences remains challenging due to the high cost of collecting preference data, training reward models, and fine-tuning. The authors proposed Contrastive Gradient Guidance (CGG) as a test-time alignment framework that eliminates the need for explicit reward models by deriving guidance signals from the contrastive difference between two diffusion models. By leveraging the gradient of the log-likelihood ratio between favored and unfavored distributions, CGG steers the diffusion process toward preference-aligned outputs during sampling. Experiments show that CGG improves preference alignment across diverse scenarios, including safety-critical and multi-preference settings, and can be combined with existing test-time alignment methods for further gains.

**Strengths:**

1. The paper is well-written, with clear motivation and logical flow throughout.
2. It presents a simple approach for performing test-time optimization across multiple preferences.

**Weaknesses:**

1. One major concern is the lack of clear justification for why the proposed method should outperform the PO-based diffusion models used during inference. Moreover, the experimental results reinforce this concern, as the observed improvements over the PO diffusion models appear marginal and not convincingly significant.

2. The connection to using multiple diffusion models for handling multiple preferences is unclear. While the idea seems motivated by the prior study (Liu et al., 2022), the paper lacks a solid theoretical explanation or justification for this linkage.

3. In practice, the method relies heavily on the use of PO-based diffusion models, introducing a strong dependency that limits its general applicability and practicality.

**Questions:**

1. If the inference process effectively performs a weighted average between the PO diffusion models and the base diffusion model, it is unclear what underlying rationale supports the expectation that this method would outperform the PO diffusion models themselves. Could you clarify this?

2. Could this method be compared with other relevant DPO-based approaches, such as Diffusion Model Alignment Using Direct Preference Optimization (Wallace et al., CVPR 2024) [1], to provide a more complete and fair experimental evaluation?

---

> ### Author Response · Authors · 2025-12-04
>
> > 1. One major concern is the lack of clear justification for why the proposed method should outperform the PO-based diffusion models used during inference. Moreover, the experimental results reinforce this concern, as the observed improvements over the PO diffusion models appear marginal and not convincingly significant.
> > - If the inference process effectively performs a weighted average between the PO diffusion models and the base diffusion model, it is unclear what underlying rationale supports the expectation that this method would outperform the PO diffusion models themselves. Could you clarify this?
> > - Could this method be compared with other relevant DPO-based approaches, such as Diffusion Model Alignment Using Direct Preference Optimization (Wallace et al., CVPR 2024) [1], to provide a more complete and fair experimental evaluation?
>
> In this work, we do not guarantee that CGG should consistently outperform the PO-based diffusion models. We leave the theoretical validation as future work.
> However, PO-based diffusion models require costly fine-tuning, with hyperparameters $\beta$ carefully tuned for the specific dataset and the reward, resulting in increased training time.
> In contrast, our method can efficiently guide pre-trained or PO-based diffusion models to adapt to the new dataset and reward during test time.
>
> > 2. The connection to using multiple diffusion models for handling multiple preferences is unclear. While the idea seems motivated by the prior study (Liu et al., 2022), the paper lacks a solid theoretical explanation or justification for this linkage.
>
> Our idea begins with the reward model, which can be interpreted as a contrastive composition of underlying favored and unfavored distributions. Therefore, the naive idea is to extend CGG for the compositional distributions version.
>
> > 3. In practice, the method relies heavily on the use of PO-based diffusion models, introducing a strong dependency that limits its general applicability and practicality.
>
> CGG framework does not rely heavily on the PO-based diffusion models. In Table 2, we also show other types of composition distributions that are not PO-based.

---

### Author Response · Authors · 2025-12-04
**General response**

**Core idea.** The reward model can be interpreted as a contrastive composition of underlying favored and unfavored distributions.

1. The contrastive compositional distributions could guide diffusion models with human preferences at test time.

**Assumption of Eq. (7) $r(c, x) = \log \frac{\rho(x \vert c)}{\kappa(x \vert c)}$.**

1. Our concept is derived from the pairwise preference data for preference alignment. In RLHF, a reward model is learned from pairwise preference data, and the likelihood of a pairwise preference data distribution is determined by the relative scores assigned by the underlying reward function.
2. Given that the pairwise preference data can naturally be decomposed into two groups: images with favored and unfavored labels by annotators, this suggests that the reward model can be interpreted more broadly as arising from two underlying distributions, a favored distribution and an unfavored distribution.
3. Motivated by the interpretation of the two underlying distributions, we assume that a relation of the form given in Eq. (7) should hold: the reward model can be seen as a contrastive composition of the favored and unfavored distributions.


**Different from existing test-time preference alignment scenarios.**, i.e., different reward models in Eq. (7) and Eq. (3).

1. Existing methods investigate test-time preference alignment with respect to the predefined reward function (trained from Eq. (3)) and fixed regularization scale $\frac{1}{\beta}$ (tuned from Eq. (4)).
2. In contrast, we begin from the perspective of favored and unfavored distributions and approximate the reward model implicitly through their contrastive relationship.
3. Consequently, rather than utilizing a predefined (explicit) reward function, CGG estimates the preference direction by reconstructing the underlying contrastive composition of the favored and unfavored distributions at test time.

**Why does CGG work for predefined reward models (PickScore, etc) for test-time preference alignment?**

1. The reason CGG also performs well under predefined reward models (such as PickScore) arises from Eq. (5). Rearranging Eq. (5) gives: $r(c, x) = \beta \log \frac{p_{\theta^\ast}(x \vert c)}{p_\mathrm{ref}(x \vert c)} + \beta \log Z(c)$ => Eq. (7). It shows that a reward model in RLHF used for test-time preference alignment can be interpreted as inducing a contrastive (density ratio) between a PO-based fine-tuned distribution and a pre-trained distribution.
2. Therefore, if we can obtain a model $p_\theta(x \vert c)$ via a PO-based method (DPO, Diffusion-DPO, DSPO, etc.), then adjusting the guidance scale in CGG allows us to approximate the test-time target reward.

---

### Meta-Review · Area_Chair_SBt7 · 2026-01-07

**Summary:**

Although the paper initially received mixed ratings (one reject, two borderline rejects, and one borderline accept), most reviewers acknowledge that the manuscript is clearly written and easy to follow (vy1u, Dq7k, i9dk), and that the proposed method, CGG, is simple yet effective, demonstrating competitive performance across multiple tasks (Dq7k, i9dk, cSmV). In the rebuttal, the authors added additional experimental results, provided clearer explanations of the core idea, clarified why CGG works with predefined reward models for test-time preference alignment (Eq. 7), discussed differences from existing test-time preference alignment settings, and improved table presentation.

However, several key issues remain unresolved. These revisions address some of the reviewers’ concerns, as confirmed through follow-up communication with Reviewers vy1u and Dq7k. vy1u maintains his score to 4, and Dq7k upgrade his score from 2 to 4. Reviewer Dq7k continues to view the proposed method as essentially a weighted combination of two distribution functions, and therefore considers the contribution and novelty to be limited. Reviewer vy1u emphasizes the lack of a solid theoretical justification raised in the initial review. Reviewer i9dk notes that in certain cases the method does not achieve state-of-the-art performance, and these limitations are not sufficiently explained. Taken together, these concerns prevent a clear consensus in favor of acceptance. Consequently, the paper is not recommended for acceptance in its current form. The authors are encouraged to further strengthen the theoretical foundations, enhance the novelty relative to existing approaches, and better analyze performance limitations before resubmission to a future venue.

**Reviewer Concerns:**

After double checking with Reviewer Dq7k through email, he appreciate that the authors added Equation (7), which addressed one of his concerns regarding the derivation of Equation (8).  However, he still has the concerns about the novelties of the proposed methods. For Reviewer i9dk, the authors have added more experimental results in the  revised paper to addressed his concerns. The authors also include more explanation of the proposed method for Reviewer cSmV and vy1u.

**Reviewer Scores:**

For Reviewers vy1u and Dq7k, I have double-checked their opinions via email, and both reviewers indicated that the rebuttal only partially addressed their concerns. In particular, Reviewer Dq7k still considers the proposed method to essentially reduce to a weighted combination of two distribution functions. As this formulation also introduces additional hyperparameter tuning, the contribution remains somewhat weak in terms of both novelty and practical impact. vy1u maintains his score while Dq7k upgrades his score from 2 to 4. For Reviewer i9dk, the authors stated in the rebuttal that the requested human evaluation would be added in future work; however, as this evaluation is not included in the current version, the reviewer is expected to keep their original score. For Reviewer cSmV, the main concerns were related to table presentation and certain experimental results, which the authors have addressed through revisions to the manuscript and a brief explanation about the result. As a result, this reviewer may potentially increase their score.

---

### Decision · Program_Chairs · 2026-01-26

Reject